

# Prevalence of physical, psychological and sexual intimate partner violence among women of reproductive age during COVID-19 in Ethiopia: a systematic review and meta-analysis

Aragaw Asfaw Hasen[1,*], Abubeker Alebachew Seid[2,*], Ahmed Adem Mohammed[2] and Kassaye Getaneh Arge[1]

[1] Department of Statistics, College of Natural and Computational Sciences, Samara University, Semera, Afar, Ethiopia
[2] Department of Nursing, College of Medicine and Health Sciences, Samara University, Semera, Afar, Ethiopia
[*] These authors contributed equally to this work.

Corresponding authors
Aragaw Asfaw Hasen,
aragawasfaw5@gmail.com
Abubeker Alebachew Seid,
abubeker2008h@gmail.com

## ABSTRACT

**Introduction.** COVID-19 preventive measures such as stay at home and isolation leads to violence against women. Intimate partner violence (IPV) is one of the common violence during this pandemic. This study aimed to assess the prevalence of physical, psychological and sexual intimate partner violence among reproductive age women during COVID-19 in Ethiopia.

**Materials and Methods.** Electronic databases such as PubMed, Google Scholar and African journals online and studies available from the occurrence of the pandemic to April 2023 were searched. Two researchers collected the data and independently performed the methodological quality assessment. To pool the collected data for each outcome with 95% confidence interval (CI), DerSimonian-Laird random effects meta-analysis was used. Publication bias was measured by Doi plot LFK index and Egger's test. Stata version 14.0 (StataCorp, College Station, Texas, USA) software was used for statistical analysis.

**Results.** A total of seven studies reported the prevalence of intimate partner violence among women in reproductive age during COVID-19, and the pooled prevalence of physical intimate partner violence was 22% (95% CI [0.12–0.32], $I^2 = 98.9\%$, $tau^2 = 0.0184$, $p < 0.001$). The pooled prevalence of psychological intimate partner violence was 28% (95% CI [0.18–0.37], $I^2 = 98.1\%$, $tau^2 = 0.0142$, $p < 0.001$). The pooled prevalence of sexual intimate partner violence was 23% (95% CI [0.13–0.34], $I^2 = 99.1\%$, $tau^2 = 0.0208$, $p < 0.001$).

**Conclusions.** During COVID-19 reproductive age women in Ethiopia were affected by intimate partner violence. Physical, psychological and sexual intimate partner violence were reported, and their prevalence was high due to the pandemic. Future studies on impact of COVID-19 on IPV among reproductive age women should be conducted in nationwide to make more comprehensive conclusion.

**PROSPERO registration number:.** CRD42023417628.

# INTRODUCTION

Among most well-known types of violence against women, one is intimate partner violence (IPV) includes physical, sexual, and emotional abuse and controlling behaviours by an intimate partner (*Donnelly, Levin & Barrett, 2021*). People in the world were restricted to remain at home, and geological disengagement during coronavirus, these measures prompts IPV among women (*Mojahed et al., 2021*). During the lockdown in South Africa, people with lower socioeconomic status were more likely to be victims of domestic violence. In order to address food insecurity during a pandemic, structural and social relief measures must be strengthened (*Mahlangu et al., 2022*). In Uganda, nearly half of the sample were facing physical IPV and reported an increase in violence during the lockdown. Women's alcohol usage was associated with four times greater odds of recent physical IPV than non users (*Miller et al., 2022*). Likewise in Egypt, the overall prevalence of economic and some types of physical and emotionally abusive behaviors have been increased after the emergence of COVID-19 pandemic (*Abu-Elenin et al., 2022*). There is a high risk that all forms of gender-based violence (GBV) will increase during the COVID-19 pandemic (*Bayu, 2020*).

COVID-19 pandemic affects the experience of intimate partner violence among adult Nigerians (*Oloniniyi et al., 2022*). Contrasted with the pandemic before coronavirus the predominance of IPV was around (10.5%), and pervasiveness of intimate partner violence during pregnancy expanded to 15.1% during the coronavirus pandemic (*Wood et al., 2022*). Especially women who are pregnant, post pregnancy, prematurely delivering, or encountering personal accomplice viciousness are at high gamble for creating emotional well-being issues during the pandemic (*Almeida et al., 2020*). Among partnered young girls and women, 27.6% experienced IPV during the pandemic in Kenya (*Decker et al., 2022*). During COVID-19 measure, large-scale lockdown aggravates family conflicts, economic distress and tension among family members (*Zhang, 2022*).

Intimate partner violence is a major public health issue, affecting one out of every three pregnant women in Ethiopia (*Belay et al., 2022*). Studies showed that the prevalence of IPV against women was enlarged during this COVID-19. It is a critical problem that is occurring all over the world for many years now, but this condition has been increased during the lockdown situation of COVID-19. So, studying the reproductive age group is critical to provide insights into the unique challenges and vulnerabilities faced by women in this age group who are experiencing intimate partner violence. Moreover, we choose this age group for several reasons. Firstly, focusing on the reproductive age group provides insights into the unique challenges and vulnerabilities faced by these women. Secondly, understanding the impact of intimate partner violence on reproductive age women can lead to more effective prevention and intervention strategies that address the intersection of violence and reproductive health. Lastly, studying this specific age group helps raise awareness about

the prevalence and impact of intimate partner violence on women's reproductive health and overall well-being, and can highlight more and better interventions that can benefit future generations. Study findings in Ethiopia (*Shitu, Yeshaneh & Abebe, 2021*; *Alemie et al., 2023*; *Fetene et al., 2022*; *Tadesse et al., 2022*; *Engda et al., 2022*; *Getinet et al., 2022*; *Gebrewahd, Gebremeskel & Tadesse, 2020*) showed that the evidence on the prevalence of intimate partner violence among reproductive age women during COVID-19 findings were not consistent.

Therefore, a comprehensive study and providing a comprehensive evidence on the prevalence of intimate partner violence among reproductive age women in Ethiopia during COVID-19 is crucial and no study reported in a summarized way. This study aimed to assess the prevalence of physical, psychological and sexual intimate partner violence among reproductive age women during COVID-19 in Ethiopia. This study provides evidence for scholars and policy makers of intimate partner violence among reproductive age women and related social problems during pandemics situations.

## Objective

This study proposed to assess the prevalence of physical, psychological and sexual intimate partner violence among reproductive women during COVID-19 in Ethiopia.

# MATERIALS AND METHODS

## Protocol registration

This study is conducted based on the Preferred Reporting Items for Systematic Reviews and Meta-Analyses (PRISMA) 2020 recommendation (*Page et al., 2021*). The protocol was registered in the International Prospective Register of Systematic Reviews, with PROSPERO registration number: CRD42023389896.

## Search strategy

A literature search was conducted in PubMed, African Journals Online and Google Scholar databases and articles published from the occurrence of the pademic to April 2023 were included. Observational studies assessed the pevalence of intimate partner violence during COVID-19 among women in reproductive age in Ethiopia were considered. Systematic searches were conducted by combining every possible predefined search terms determined by Medical Subject Headings (MeSH) and Keywords. The systematic procedure verifies the literature search encompasses all published studies on the impact of COVID-19 on the prevalence of intimate partner violence among reproductive age women in Ethiopia. The duplicates from the search results were removed using Mendeley (*Kwon et al., 2015*).

Two researchers (AAH and AAS) separately screened titles and abstracts of the studies, and any disagreement between the researchers was hundled by discussion with another researcher (AAM and KGA). The search was performed using keywords: "magnitude", "prevalence", "domestic violence", "psychological violence", "intimate partner violence", "violence against women", "gender-based violence", "sexual violence", "physical violence", "emotional violence", "SARS- CoV-2", "COVID-19" "pregnant women","prenatal", "perinatal", "postpartum", "antenatal", "postnatal", "peurperal",

**Table 1  PubMed search strategy.**

| Search number | Search detail |
|---|---|
| #1 | "COVID-19"[MeSH Terms] |
| #2 | "intimate partner violence"[Mesh Terms] |
| #3 | "pregnancy" [Mesh Terms] |
| #4 | "COVID-19"[Title/Abstract] OR "2019 novel coronavirus disease"[Title/Abstract] OR "2019 novel coronavirus infection"[Title/Abstract] OR "2019 ncov disease"[Title/Abstract] OR "2019 ncov infection"[Title/Abstract] OR "covid 19 pandemic"[Title/Abstract] OR "covid 19 pandemics" [Title/Abstract] OR "covid 19 virus disease"[Title/Abstract] OR "covid 19 virus infection"[Title/Abstract] OR "COVID19"[Title/Abstract] OR "coronavirus disease 2019"[Title/Abstract] OR "coronavirus disease 19"[Title/Abstract] OR "sars coronavirus 2 infection"[Title/Abstract] OR "sars cov 2 infection"[Title/Abstract] OR "severe acute respiratory syndrome coronavirus 2 infection" [Title/Abstract] OR "SARS-CoV-2" [Title/Abstract] OR "2019 novel coronavirus" [Title/Abstract] OR "2019 novel coronavirus"[Title/Abstract] OR "2019- nCoV"[Title/Abstract] OR "covid 19 virus"[Title/Abstract] OR "covid19 virus"[Title/Abstract] OR"Coronavirus disease 2019 virus"[Title/Abstract] OR "SARS coronavirus 2" [Title/Abstract] OR "SARS cov 2 virus"[Title/Abstract] OR "severe acute respiratory syndrome coronavirus 2"[Title/Abstract] OR "Wuhan coronavirus"[Title/Abstract] OR "Wuhan seafood market pneumonia virus"[Title/Abstract] |
| #5 | "domestic violence" [Title/Abstract] OR "psychological violence" [Title/Abstract] OR "intimate partner violence" [Title/Abstract] OR "violence against women," [Title/Abstract] OR "gender based violence" [Title/Abstract] OR "sexual violence" [Title/Abstract] OR "physical violence" [Title/Abstract] OR "emotional violence" [Title/Abstract] |
| #6 | "Women"[Title/Abstract] OR "pregnant women"[Title/Abstract] OR "prenatal"[Title/Abstract] OR " perinatal"[Title/Abstract] OR "postpartum" [Title/Abstract] OR "antenatal" [Title/Abstract] OR "postnatal" [Title/Abstract] OR "peurperal" [Title/Abstract] OR "lactating women" [Title/Abstract] OR "reproductive age women"[Title/Abstract] OR "child bearing women" [Title/Abstract] AND "Ethiopia"[Title/Abstract] |
| #7 | #1 OR #4 |
| #8 | #2 OR #5 |
| #9 | #3 OR #6 |
| #10 | #7 AND #8 AND #9 |

"lactating women", "reproductive age women" ,"child bearing women" ,"Ethiopia". We also used Boolean operators "AND" and "OR". All electronic sources of information were searched for the articles conducted up to April 2023. The search strategy of PubMed database is presented in Table 1.

## Eligibility criteria
### Inclusion criteria

For this study only observational studies that examined the prevalence of common types of intimate partner violence among reproductive age women during COVID-19 pandemic in Ethiopia. This study is employed following the condition, context and population (CoCoPop) framework to ease the searching strategy and organization of search terms.
*Condition*: Intimate partner violence.
*Context*: During the COVID -19 pandemic in Ethiopia.

*Population*: All reproductive age women.
*Study design*: Observational studies.

### Exclusion criteria

The following types of studies were excluded:
1. Studies emphasis on whole population.
2. Studies that did not have sufficient statistical data to be extracted.
3. Randomized controlled trials, systematic review and meta-analysis, editorials, conference abstracts and opinions were excluded.

## Outcome measures

The main outcome of this study is the pooled prevalence of physical, psychological and sexual intimate partner violence among reproductive age women during COVID-19 in Ethiopia. It can be measured by percentage and inline with its 95% confidence interval (CI).

## Selection of studies

Two researchers (AAH and AAS) appraised the studies based on eligibility criteria. Firstly, the authors evaluated both the titles and abstracts of the studies from the searched databases. Full-text screening was done to screen the full texts. Also, we have a rationale for inclusion and exclusion of studies in the PRISMA diagram. Lastly, the final list of studies for data extraction for systematic review and meta-analysis was set for the analysis.

## Data extraction

The data extraction was done by two researchers (AAH and KGA) independently. There was pre-test the data extraction form on two preliminary surveyed studies, to ease the collection of all necessary data required for the systematic review and meta-analysis. Disagreements were handled by discussion. Data were collected from primary studies. Specifically, region, study design, number of cases, sample size, sampling design/ method, instrument/tools, study population, average age and prevalence of intimate partner violence types with their prevalence.

## Risk of bias assessment

Two researchers (AAH and AAM) separately weighed the quality of the included studies by the Newcastle-Ottawa Scale (NOS) (*Peterson et al., 2011*). According to this scale, 3 parameters named selection, comparability, and assessment of exposure/outcome were considered. Studies with less than 5 scores were considered low quality, 5–7 scores of moderate quality, and more than 7 scores of high quality (*Ssentongo et al., 2020*). Studies with moderate and above quality score were considered for this study.

## Data synthesis

The extracted data is imported to Stata version 14.0 (StataCorp, College Station, Texas, USA) software to conduct the meta-analysis. To pool the raw data for each outcome with 95% confidence interval (CI), DerSimonian-Laird random-effects meta-analysis is used. Assessment of heterogeneity was checked by $I^2$ and Cochran's *Q*-statistic (*Bowden et al.,*

*2011*; *Zhu et al., 2020*). Subgroup analyses was performed by regions to decide the source of heterogeneity. Publication bias was checked by DOI plot Luis Furuya Kanamori (LFK) index and Egger's test (*Furuya-Kanamori, Barendregt & Doi, 2018*; *Furuya-Kanamori & Doi, 2021*). According to the LFK index, a value outside the interval -1 and 1 were considered as asymmetry (*i.e.,* publication bias) (*Furuya-Kanamori & Doi, 2021*).

**Patient and public involvement**
No patient and public involvement.

# RESULTS

A PRISMA diagram proving the steps of database search and filtering process for the study on prevalence of intimate partner violence among reproductive age women during the COVID-19 pandemic was shown in Fig. 1. From databases search, initially 19 studies were identified. Due to duplication, six studies were removed. A total of 13 studies were screened and full text studies were examined, one study was excluded by its title and abstract. Five studies were removed by reasons that did not met inclusion criteria such as studies were not observational and the studies were not considered the COVID-19 pandemic period. Finally, we documented seven studies fitting to this systematic review and meta-analysis.

## Study characteristics
In this comprehensive study, a total of 4,439 samples were included to assess the prevalence of physical, psychological, and sexual intimate partner violence among women of reproductive age during COVID-19 in Ethiopia. These samples were collected from Ethiopia, including various regions *i.e.,* four studies from Amhara region (*Alemie et al., 2023*; *Tadesse et al., 2022*; *Engda et al., 2022*; *Getinet et al., 2022*), one study from Tigray region (*Gebrewahd, Gebremeskel & Tadesse, 2020*), one study from SNNP region -(*Shitu, Yeshaneh & Abebe, 2021*) and one study from South West Ethiopia region (*Fetene et al., 2022*). Among these, cases of physical, psychological, and sexual intimate partner violence were reported as 988, 1,233, and 1,020 respectively. In this systematic review and meta-analysis, we included seven cross-sectional studies (*Shitu, Yeshaneh & Abebe, 2021*; *Alemie et al., 2023*; *Fetene et al., 2022*; *Tadesse et al., 2022*; *Engda et al., 2022*; *Getinet et al., 2022*; *Gebrewahd, Gebremeskel & Tadesse, 2020*) on the prevalence of IPV among reproductive age women during COVID-19 pandemic in Ethiopia. The summarized data of the key characteristics of the included studies were showed in Table 2.

## Risk of bias assessment
The risk of bias of included studies was assessed by using the modified Newcastle Ottawa scale (NOS) for cross-sectional studies quality assessment showed in Table 1. We cosidered moderate quality, and high quality studies for this review. Accordingly, one study (*Shitu, Yeshaneh & Abebe, 2021*) was appraised as moderate quality and six studies (*Alemie et al., 2023*; *Fetene et al., 2022*; *Tadesse et al., 2022*; *Engda et al., 2022*; *Getinet et al., 2022*; *Gebrewahd, Gebremeskel & Tadesse, 2020*) were appraised as high quality and included for final systematic review and meta analysis.
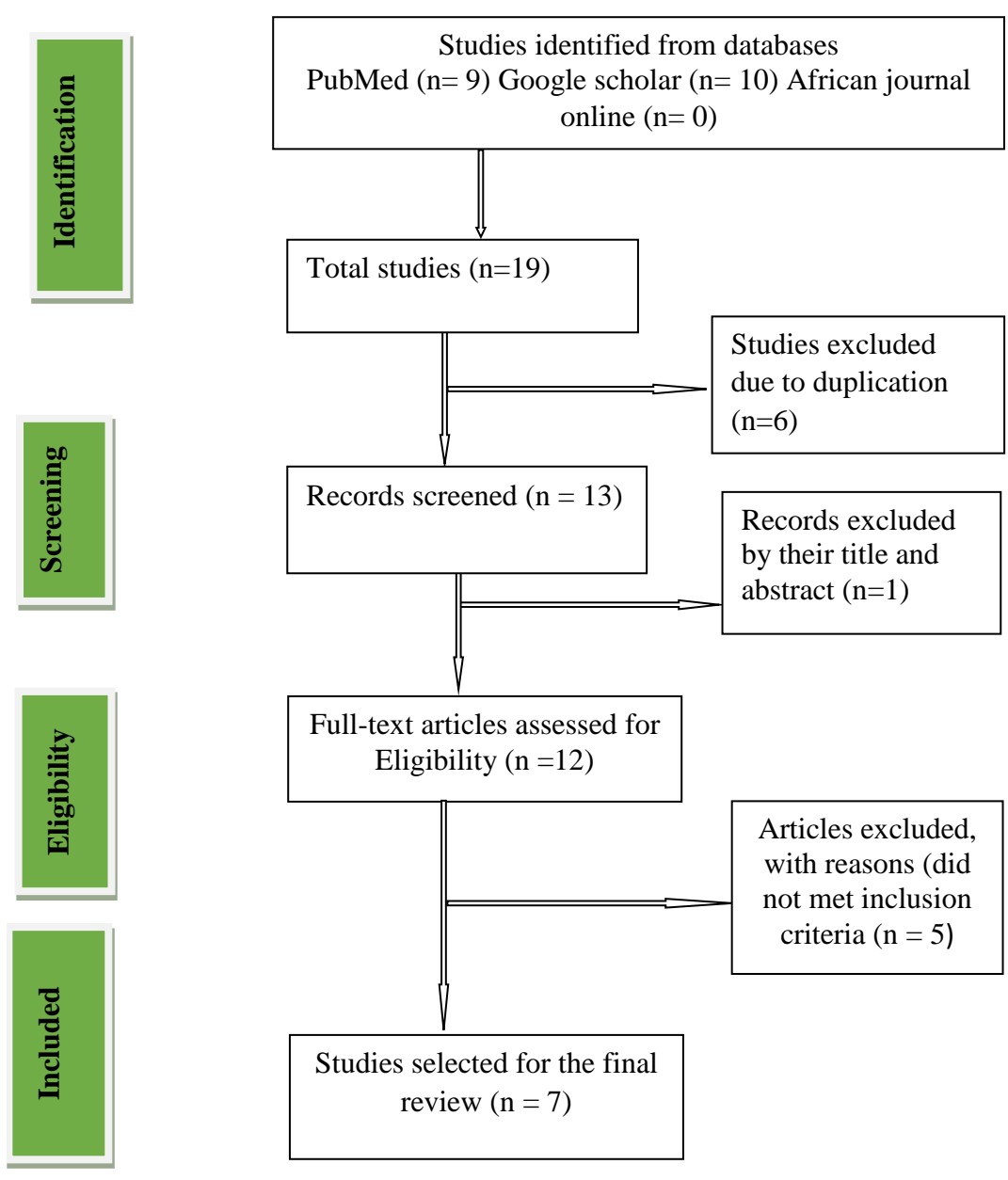

**Figure 1** Preferred reporting items for systematic reviews and meta-analyses (PRISMA) flow chart.

## Publication bias

Publication bias was assessed by the DOI plot (*Furuya-Kanamori, Barendregt & Doi, 2018*), a tool used to visualize asymmetry and by the LFK index (*Furuya-Kanamori & Doi, 2021*), a tool used to detect and quantify asymmetry of study effects. As showed in Fig. 2, for physical intimate partner violence (LFK index = 0.99, Egger's test $p$ value =0.15), for psychological intimate partner violence (LFK index = 0.98, Egger's test $p$ value =0.12), for sexual intimate partner violence (LFK index = 0.99, Egger's test $p$ value =0.11). From these

results LFK index is between the interval [−1 and 1] indicates that there is no asymmetry, similarly the Egger's test $p$ value are not statisticaly signficant ($p$ value >0.05) support that no publication bias was observed in studies included for this systematic review and mata analysis.

## Meta analysis results

### Pooled prevalence of physical intimate partner violence

A total of seven studies reported the prevalence of physical intimate partner violence among women in reproductive age during COVID-19, and the pooled prevalence of physical intimate partner violence was 22% (95% CI [0.12–0.32], $I^2 = 98.9\%$, $tau^2 = 0.0184$, $p < 0.001$). As shown in Fig. 3, there is considerable heterogeneity among study findings on prevalence of physical intimate partner violence among women in reproductive age through COVID-19 pandemic in Ethiopia.

### Subgroup analysis of physical IPV by region

To evaluate the sources of the heterogeneity we peformed the subgroup analysis by region accordingly the prevalence of physical intimate partner violence among women in reproductive age during the COVID-19 pandemic in Ethiopia was not homogeneous across regions ($Q = 103.53$, $df = 3$, $p < 0.001$) (Table 3). The prevalence of physical IPV in Amhara, SNNP, South West Ethiopia and Tigray was 25%, 18%, 30% and 9% respectively (Fig. 4). The prevalence was higher in South West Ethiopia followed by Amhara and SNNP. This variance might be due to the differences in: community level of awareness regarding physical IPV during COVID-19, accessibility of information on gender-based issues and legal implementation across regions, moreover the prevalence of intimate partner violence in Ethiopia is likely influenced by a combination of factors including educational levels diffrences, income level, cultural norms, access to support services, and urban $vs$ rural disparities.

### Pooled prevalence of psychological intimate partner violence

A total of seven studies reported the prevalence of psychological intimate partner violence among women in reproductive age during COVID-19, and the pooled prevalence of intimate partner violence was 28% (95% CI [0.19–0.37], $I^2 = 98.1\%$, $tau^2 = 0.0142$, $p < 0.001$). As shown in Fig. 5, there is considerable heterogeneity among study findings on prevalence of psychological intimate partner violence among women in reproductive age through COVID-19 pandemic in Ethiopia.

### Subgroup analysis of psychological IPV by region

To assess the sources of differences we peformed the subgroup analysis by region. Accordingly the prevalence of psychological IPV in Amhara, SNNP, South West Ethiopia and Tigray was 31%, 36%, 22% and 13% respectively (Fig. 6). The prevalence was higher in SNNP, followed by Amhara and South West Ethiopia. The prevalence of psychological intimate partner violence among women in reproductive age during the COVID-19 pandemic in Ethiopia is not consistent across regions ($Q = 81.26$, $df = 3$, $p < 0.001$). This

Peer J

**Table 2** **Study characteristics and quality of the included studies for meta analysis of IPV among reproductive age women during COVID-19 pandemic in Ethiopia.**

| No | Authors | Year | Region | Population | Study design | Sampling method | Types of IPV | n | Cases of IPV | Instrument | Age (average) | Quality |
|---|---|---|---|---|---|---|---|---|---|---|---|---|
| 1 | Alemie et al. | 2023 | Amhara | HIV Positive Women | CS | Systematic sampling | Sexual | 626 | 320 | WHO's multi-country study and the EDHS on women's health and domestic violence tool | 32.78 | 8 |
| | | | | | | | Physical | 626 | 343 | | | |
| | | | | | | | Psychological | 626 | 306 | | | |
| 2 | Getinet et al. | 2022 | Amhara | Reproductive age women | CS | Multistage sampling | Sexual | 804 | 226 | Women's Abuse Screening Test | 32 | 8 |
| | | | | | | | Physical | 804 | 188 | | | |
| | | | | | | | Psychological | 804 | 286 | | | |
| 3 | Shitu et al. | 2021 | SNNP | Reproductive age women | CS | All Pregnant women/census | Sexual | 448 | 133 | WHO core questionnaire on domestic intimate partner violence | 26.5 | 7 |
| | | | | | | | Physical | 448 | 82 | | | |
| | | | | | | | Psychological | 448 | 162 | | | |
| 4 | Engda et al. | 2022 | Amhara | All child bearing women | CS | Multistage sampling | Sexual | 700 | 66 | Eight items of women abuse screening tool | 33.04 | 8 |
| | | | | | | | Physical | 700 | 76 | | | |
| | | | | | | | Psychological | 700 | 139 | | | |
| 5 | Fetene et al. | 2022 | S.West | Pregnant women | CS | Systematic sampling | Sexual | 590 | 158 | Tools for the assessment of domestic violence against women in low-income country settings | 31.9 | 8 |
| | | | | | | | Physical | 590 | 176 | | | |
| | | | | | | | Psychological | 590 | 131 | | | |
| 6 | Tadesse et al. | 2022 | Amhara | Married women | CS | Systematic sampling | Sexual | 589 | 81 | Tools for the assessment of domestic violence against women in low-income country settings | 32 | 8 |
| | | | | | | | Physical | 589 | 65 | | | |
| | | | | | | | Psychological | 589 | 118 | | | |
| 7 | Gebrewahd et al. | 2020 | Tigray | Reproductive age women | CS | Systematic sampling | Sexual | 682 | 36 | WHO core questionnaire on domestic intimate partner violence | 29.79 | 8 |
| | | | | | | | Physical | 682 | 58 | | | |
| | | | | | | | Psychological | 682 | 91 | | | |

**Notes.**

SNNP, Southern Nations Nationalities and People; S.West, South West Ethiopia; CS, Cross-sectional; n, sample size; IPV, Intimate Partner Violence.

(*Alemie et al., 2023*; *Getinet et al., 2022*; *Shitu, Yeshaneh & Abebe, 2021*; *Engda et al., 2022*; *Fetene et al., 2022*; *Tadesse et al., 2022*; *Gebrewahd, Gebremeskel & Tadesse, 2020*).

A. Physical IPV studies

B. Psychological IPV studies

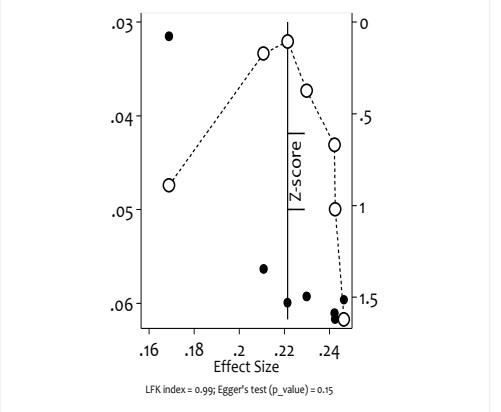

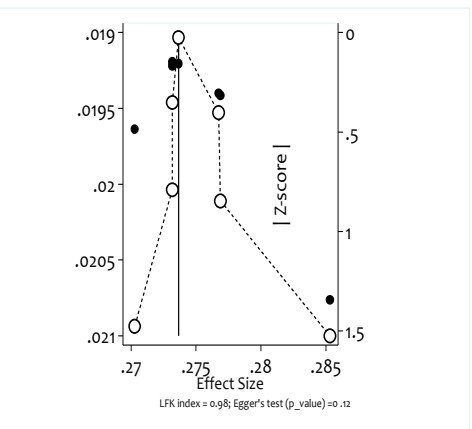

c. Sexual IPV studies

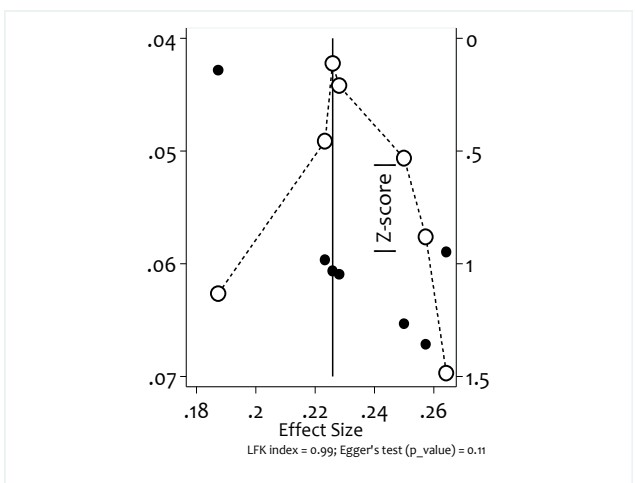

**Figure 2** **Assessment of publication bias of included studies using Doi plot and LFK index.**

difference might be due to the disparity in awareness regarding psychological IPV during COVID-19, ease of access of information on gender-based issues, also a combination of factors including educational levels differences, income level, cultural norms, access to support services, and urban *vs.* rural disparities might be the source of heterogeneity.

## Pooled prevalence of sexual intimate partner violence

A total of seven studies reported the prevalence of sexual intimate partner violence among women in reproductive age during COVID-19, and the pooled prevalence of sexual intimate partner violence was 23% (95% CI [0.13–0.34], $I^2 = 99.1\%$, $tau^2 = 0.0208$, $p < 0.001$). As shown in Fig. 7, there is considerable heterogeneity among study findings on prevalence of sexual intimate partner violence among women in reproductive age through COVID-19 pandemic in Ethiopia.

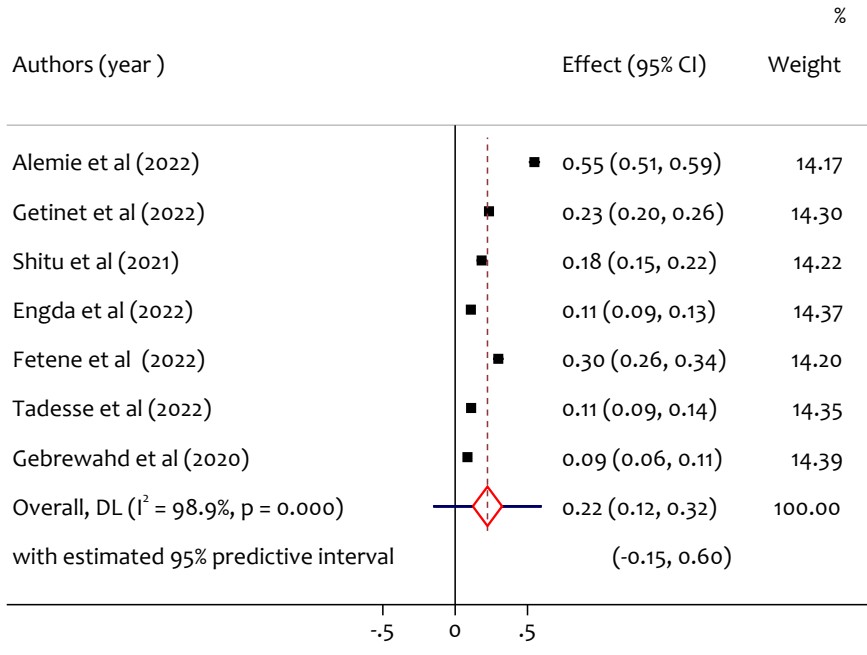

**Figure 3** Forest plot for the prevalence of physical IPV among reproductive age women during COVID-19 pandemic in Ethiopia. (*Alemie et al., 2023*; *Getinet et al., 2022*; *Shitu, Yeshaneh & Abebe, 2021*; *Engda et al., 2022*; *Fetene et al., 2022*; *Tadesse et al., 2022*; *Gebrewahd, Gebremeskel & Tadesse, 2020*).

**Table 3  Type of intimate partner violence and assessment of heterogenity between subgroups.**

| Type of intimate partner violence | Subgroup | Cochran's Q-statistic | Df | P value |
|---|---|---|---|---|
| Physical | Region | 103.53 | 3 | $p < 0.001$ |
| Psychological | Region | 81.26 | 3 | $p < 0.001$ |
| Sexual | Region | 196.89 | 3 | $p < 0.001$ |

Notes.
Df, degree of freedom.

## Subgroup analysis of sexual IPV by region

We peformed the subgroup analysis by region to assess the sources of heterogeneity among studies. Accordingly the prevalence of sexual intimate partner violence among women in reproductive age during the COVID-19 pandemic in Ethiopia is not homogeneous across regions ($Q = 196.89$, $df = 3$, $p < 0.0001$) (Table 3). The prevalence of of sexual IPV in Amhara, SNNP, South West Ethiopia and Tigray was 26%, 30%, 27% and 5% respectively (Fig. 8). The prevalence of sexual IPV was higher in SNNP, Amhara and South West Ethiopia compared to Tigray. This difference might be due to the disparity on factors such as educational levels difrences, income level, cultural norms, access to support services, and residence might be the source of heterogeneity.

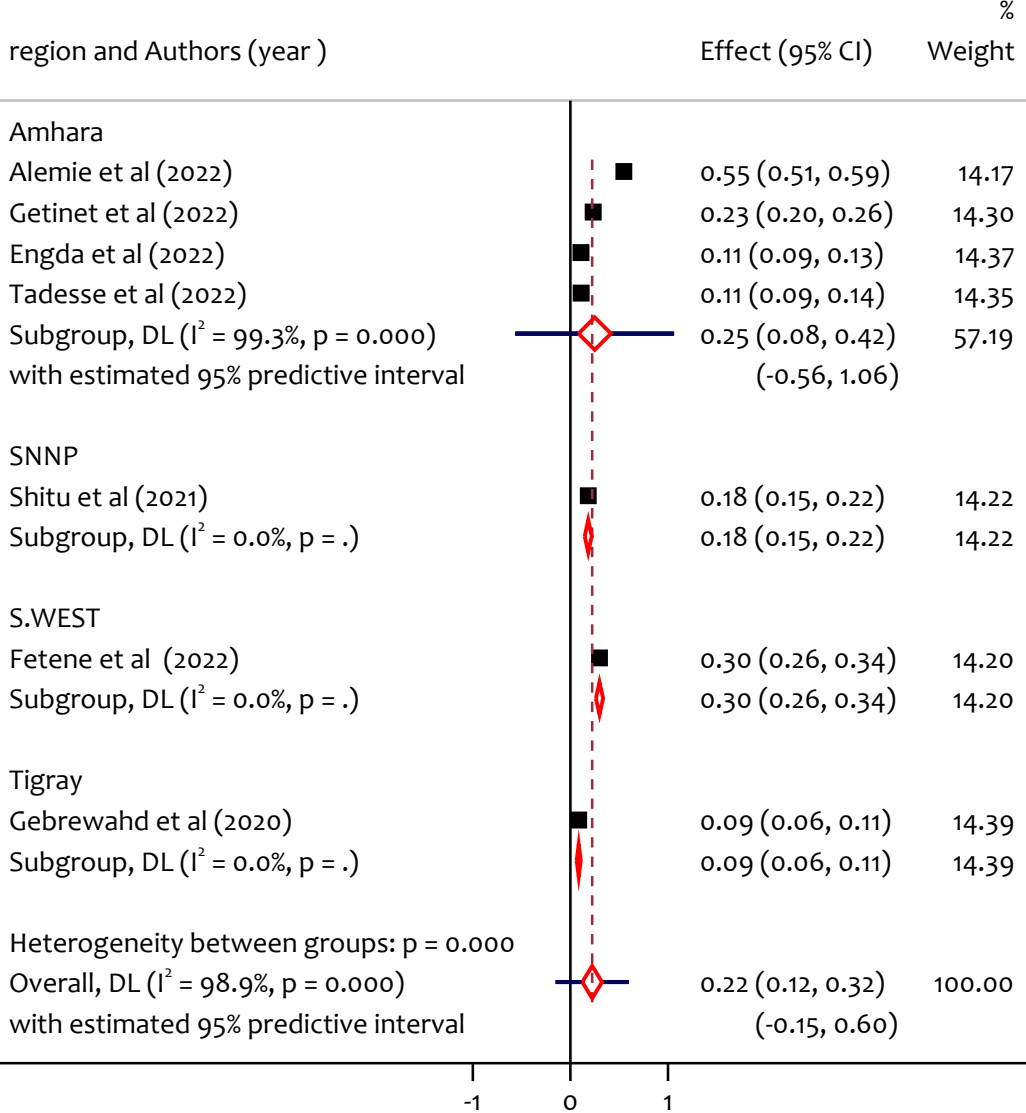

| region and Authors (year ) | Effect (95% CI) | %<br>Weight |
|---|---|---|
| **Amhara** | | |
| Alemie et al (2022) | 0.55 (0.51, 0.59) | 14.17 |
| Getinet et al (2022) | 0.23 (0.20, 0.26) | 14.30 |
| Engda et al (2022) | 0.11 (0.09, 0.13) | 14.37 |
| Tadesse et al (2022) | 0.11 (0.09, 0.14) | 14.35 |
| Subgroup, DL ($I^2$ = 99.3%, p = 0.000) | 0.25 (0.08, 0.42) | 57.19 |
| with estimated 95% predictive interval | (-0.56, 1.06) | |
| | | |
| **SNNP** | | |
| Shitu et al (2021) | 0.18 (0.15, 0.22) | 14.22 |
| Subgroup, DL ($I^2$ = 0.0%, p = .) | 0.18 (0.15, 0.22) | 14.22 |
| | | |
| **S.WEST** | | |
| Fetene et al  (2022) | 0.30 (0.26, 0.34) | 14.20 |
| Subgroup, DL ($I^2$ = 0.0%, p = .) | 0.30 (0.26, 0.34) | 14.20 |
| | | |
| **Tigray** | | |
| Gebrewahd et al (2020) | 0.09 (0.06, 0.11) | 14.39 |
| Subgroup, DL ($I^2$ = 0.0%, p = .) | 0.09 (0.06, 0.11) | 14.39 |
| | | |
| Heterogeneity between groups: p = 0.000 | | |
| Overall, DL ($I^2$ = 98.9%, p = 0.000) | 0.22 (0.12, 0.32) | 100.00 |
| with estimated 95% predictive interval | (-0.15, 0.60) | |

NOTE: Weights and between-subgroup heterogeneity test are from random-effects model

**Figure 4** Subgroup analysis of prevalence of physical IPV among reproductive age women during COVID-19 pandemic by region. (*Alemie et al., 2023*; *Getinet et al., 2022*; *Engda et al., 2022*; *Tadesse et al., 2022*; *Shitu, Yeshaneh & Abebe, 2021*; *Fetene et al., 2022*; *Gebrewahd, Gebremeskel & Tadesse, 2020*).

# DISCUSSION

In this comprehensive study we have shown that due to COVID-19 pandemic physical, psychological and sexual intimate violence among reproductive age women was increased in Ethiopia. To the best of our knowledge, this systematic review and meta-analysis is the first of its kind that assessed the pooled prevalence physical, psychological and sexual intimate partner violence during the COVID-19 pandemic in Ethiopia. We focused specifically on physical, psychological, and sexual intimate partner violence among women of reproductive
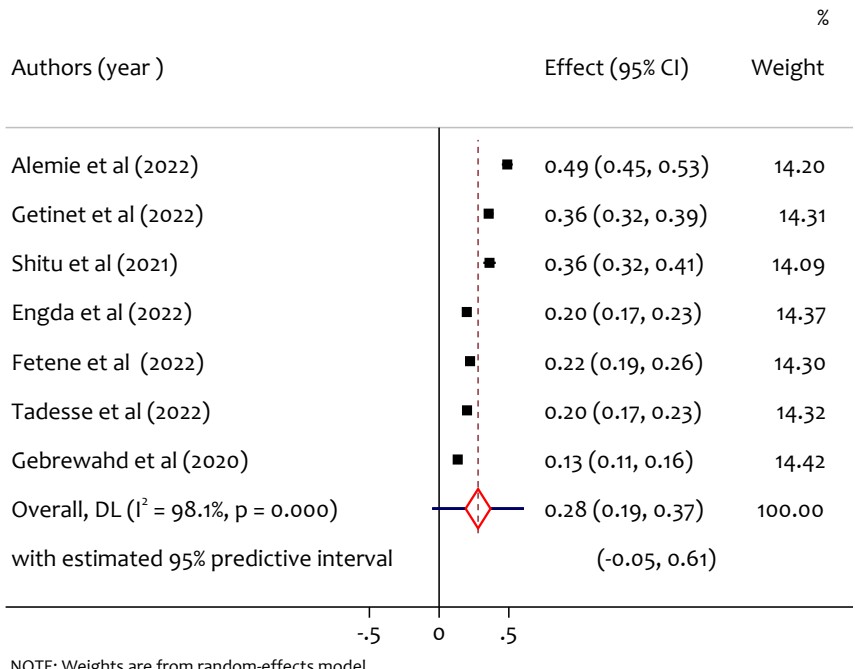

%

| Authors (year) | Effect (95% CI) | Weight |
|---|---|---|
| Alemie et al (2022) | 0.49 (0.45, 0.53) | 14.20 |
| Getinet et al (2022) | 0.36 (0.32, 0.39) | 14.31 |
| Shitu et al (2021) | 0.36 (0.32, 0.41) | 14.09 |
| Engda et al (2022) | 0.20 (0.17, 0.23) | 14.37 |
| Fetene et al (2022) | 0.22 (0.19, 0.26) | 14.30 |
| Tadesse et al (2022) | 0.20 (0.17, 0.23) | 14.32 |
| Gebrewahd et al (2020) | 0.13 (0.11, 0.16) | 14.42 |
| Overall, DL ($I^2$ = 98.1%, p = 0.000) | 0.28 (0.19, 0.37) | 100.00 |
| with estimated 95% predictive interval | (-0.05, 0.61) | |

-.5    0    .5

NOTE: Weights are from random-effects model

**Figure 5** Forest plot for the prevalence of psychological IPV among reproductive age women during COVID-19 pandemic in Ethiopia. (*Alemie et al., 2023*; *Getinet et al., 2022*; *Shitu, Yeshaneh & Abebe, 2021*; *Engda et al., 2022*; *Fetene et al., 2022*; *Tadesse et al., 2022*; *Gebrewahd, Gebremeskel & Tadesse, 2020*).

age during COVID-19 in Ethiopia. We choose this age group for several reasons. Firstly, focusing on the reproductive age group provides insights into the unique challenges and vulnerabilities faced by these women. Secondly, understanding the impact of intimate partner violence on reproductive age women can lead to more effective prevention and intervention strategies that address the intersection of violence and reproductive health. Lastly, studying this specific age group helps raise awareness about the prevalence and impact of intimate partner violence on women's reproductive health and overall well-being, and can highlight more and better interventions that can benefit future generations. This study includes seven articles focusing on the prevalence physical, psychological and sexual intimate partner violence among reproductive age women during COVID-19 in Ethiopia. We believed that all of the included studies are conducted with the ethical guideline. The pooled prevalence of intimate partner violences were discussed.

A total of seven studies reported the prevalence of physical intimate partner violence among women in reproductive age during COVID-19, and the pooled prevalence of physical intimate partner violence was 22% (95% CI [0.12–0.32]). The result is higher than in Bangladesh prevalence of physical intimate partner violence was 15.29% (*Rayhan & Akter, 2021*) and lower than in Egypt, physical intimate partner violence prevalence was (38.9%) (*Elsaid et al., 2022*). Compared to result of study in Ethiopia before COVID-19, the prevalence was 16% (*Alebel et al., 2018*), this indicate the pandemic rises the risk of physical IPV among women in Ethiopia. The reason might be during the COVID-19 pandemic

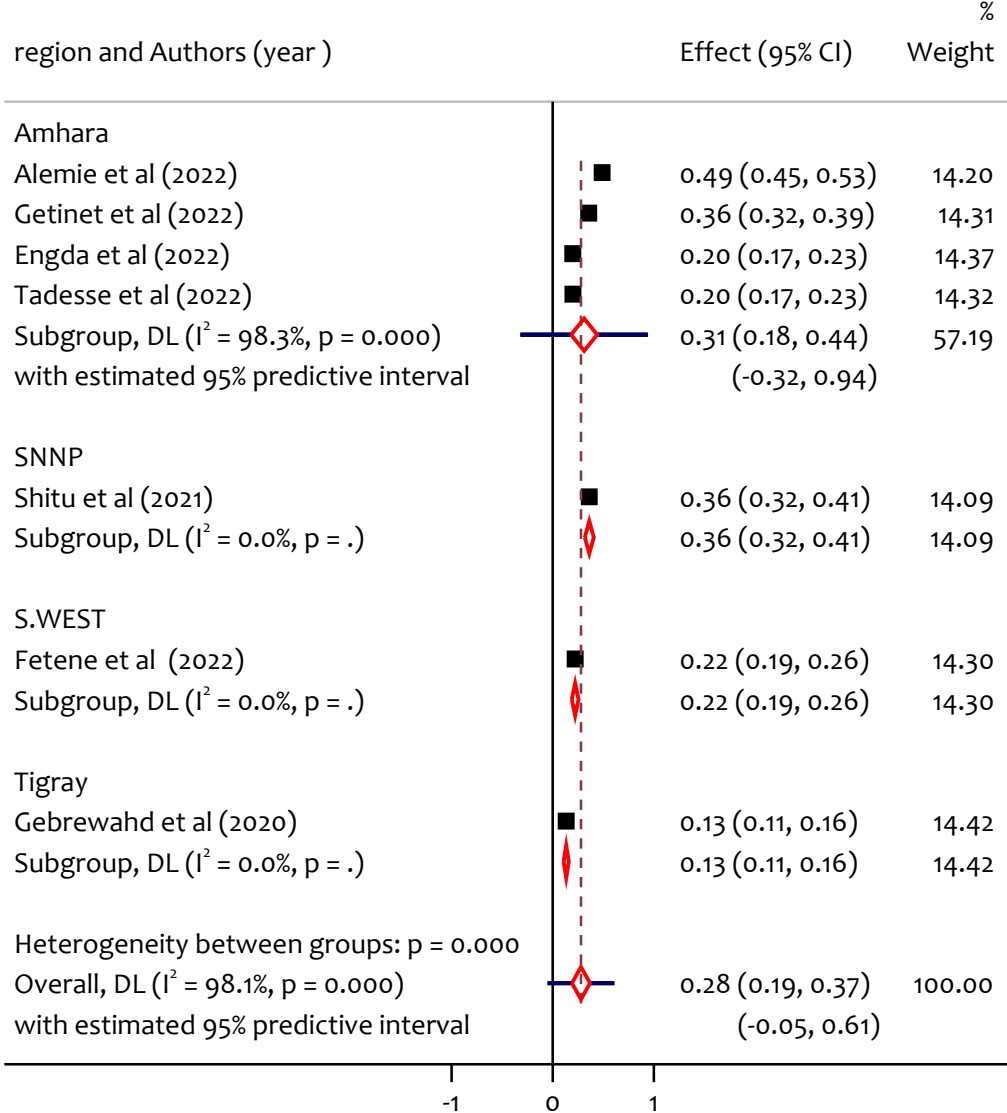

Figure 6 Subgroup analysis of prevalence of psychological IPV among reproductive age women during COVID-19 pandemic by region. (*Alemie et al., 2023*; *Getinet et al., 2022*; *Engda et al., 2022*; *Tadesse et al., 2022*; *Shitu, Yeshaneh & Abebe, 2021*; *Fetene et al., 2022*; *Gebrewahd, Gebremeskel & Tadesse, 2020*).

locked down and stay at home COVID-19 prevention measures leading to physical intimate partner violence.

Similarly seven studies findings were pooled to report the prevalence of psychological intimate partner violence among women in reproductive age during COVID-19, and the pooled prevalence was 28% (95% CI [0.19–0.37]). The result is lower than in Bangladesh prevalence of psychological intimate partner violence was 44.12% (*Rayhan & Akter, 2021*) and in Egypt emotional violence was the most prevalent (43.5%) (*Elsaid et al., 2022*). Likewise when we compare this result with study in Ethiopia before COVID-19 (*Alebel et*

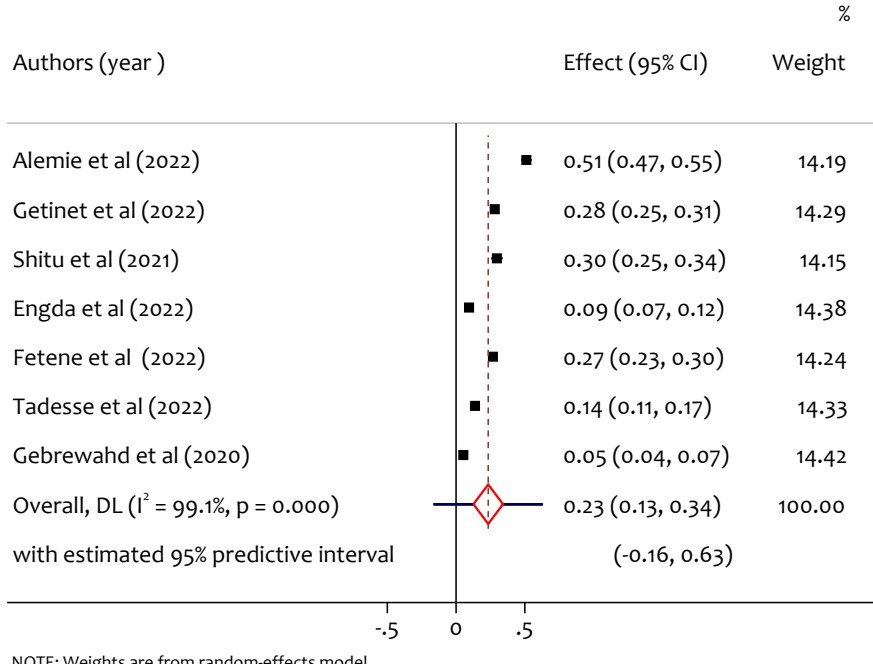

**Figure 7** Forest plot for the prevalence of sexual IPV among reproductive age women during COVID-19 pandemic in Ethiopia. (*Alemie et al., 2023*; *Getinet et al., 2022*; *Shitu, Yeshaneh & Abebe, 2021*; *Engda et al., 2022*; *Fetene et al., 2022*; *Tadesse et al., 2022*; *Gebrewahd, Gebremeskel & Tadesse, 2020*).

*al., 2018*), the prevalence was 21%, and indeed the pandemic increases the prevalence of psychological intimate partner violence among women in Ethiopia.

In this meta analysis we also reported the prevalence of sexual intimate partner violence among women in reproductive age during COVID-19, and a total of seven studies the pooled prevalence of sexual intimate partner violence was 23% (95% CI [0.13–0.34]). The result is higher than in Bangladesh, the prevalence of sexual intimate partner violence was 10.59% (*Rayhan & Akter, 2021*) and in Egypt prevalence was 17.5% (*Elsaid et al., 2022*). Also in Ethiopia study conducted before COVID-19 pandemic reported that the prevalence of sexual intimate violence was 12% (*Alebel et al., 2018*). This supports the pandemic rises the risk of sexual intimate partner violence among reproductive age women in Ethiopia.

The finding of this comprehensive study can provide valuable insights in to prevalence and impact of intimate partner violence during COVID-19 in Ethiopia. This can help healthcare providers, policy makers and researchers on the challenges faced by idividuals experiencing intimate partner violence during COVID-19 and develop timely intervention to address their needs. Moreover, this study can also shed light on the effectiveness of the existing support services and the gaps needs to be adressed to better protect and support survivors of intimate partner violence.

Whereas, this study is with strengths and limitation. This is the first systematic review and meta-analysis to examine the pooled prevalence of physical, psychological, and sexual intimate partner violence among reprodutive age women during COVID-19 in

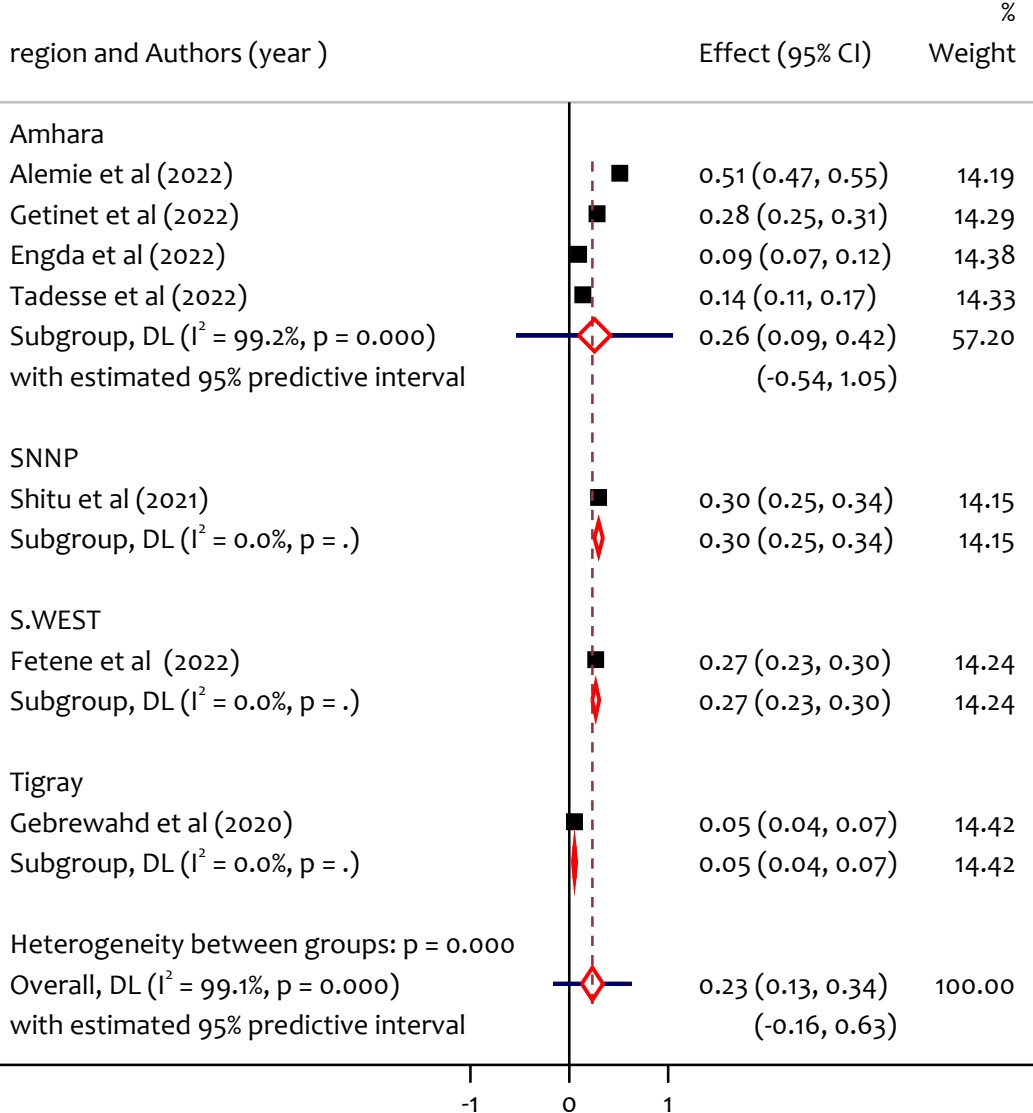

**Figure 8  Subgroup analysis of prevalence of sexual IPV among reproductive age women during COVID-19 pandemic by region.** (*Alemie et al., 2023*; *Getinet et al., 2022*; *Engda et al., 2022*; *Tadesse et al., 2022*; *Shitu, Yeshaneh & Abebe, 2021*; *Fetene et al., 2022*; *Gebrewahd, Gebremeskel & Tadesse, 2020*).

Ethiopia. Searching, screening, data extraction and risk of bias assessment were done by two researchers independently. The Newcastle-Ottawa Scale was used to assess the quality of included studies. The limited number of studies in Ethiopia on the impact of COVID-19 on prevalence of intimate partner violence of women in reproductive age, it may worsen the generalization of the overall prevalence of intimate partner violence among women in reproductive age in Ethiopia.

## CONCLUSION

During the COVID-19 pandemic the prevalence of physical, psychological and sexual intimate partner violence among women in reproductive age was high in Ethiopia. This review helped to highlight the need of interventions on intimate partner violence that can be taken during COVID-19. In addressing vulnerability for intimate partner violence among reproductive age women, education on various forms of intimate partner violence issues affecting women and girls especially during pandemic situations is vital. The analysis underscores the importance of multidisciplinary collaboration involving healthcare providers, socia lworkers, law enforcement agencies and policy makers to effectively combat intimate partner violence and protect the reproductive health and overall wellbeing of women in their reproductive age. Mental health counseling and life-skill improvement programmes, community mobilization and behavioral change communication, empower literacy rate in the family and economy, reform legal frameworks might reduces intimate partner violence. Future studies should be conducted in nationwide to make more comprehensive conclusion on impact of COVID-19 on intimate partner violence among reproductive age women in Ethiopia.

### Abbreviations

| | |
|---|---|
| AOR | Adjusted odds ratio |
| CI | Confidence Interval |
| GBV | Gender-Based Violence |
| IPV | Intimate Partner Violence |
| MeSH | Medical Subject Headings |
| NOS | Newcastle Ottawa Quality Assessment Scale |
| PRISMA | Preferred Reporting Items for Systematic Review and Meta-Analysis |

### Funding

The authors received no funding for this work.

### Competing Interests

The authors declare there are no competing interests.

### Author Contributions

- Aragaw Asfaw Hasen conceived and designed the experiments, performed the experiments, analyzed the data, prepared figures and/or tables, authored or reviewed drafts of the article, and approved the final draft.
- Abubeker Alebachew Seid conceived and designed the experiments, performed the experiments, analyzed the data, prepared figures and/or tables, authored or reviewed drafts of the article, and approved the final draft.
- Ahmed Adem Mohammed conceived and designed the experiments, performed the experiments, analyzed the data, authored or reviewed drafts of the article, and approved the final draft.

- Kassaye Getaneh Arge performed the experiments, analyzed the data, prepared figures and/or tables, authored or reviewed drafts of the article, and approved the final draft.

## Data Availability

This is a systematic review/meta-analysis.

## Supplemental Information

Supplemental information for this article can be found online at http://dx.doi.org/10.7717/peerj.17812#supplemental-information.

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
