# Peer review of "Prevalence of physical, psychological and sexual intimate partner violence among women of reproductive age during COVID-19 in Ethiopia: a systematic review and meta-analysis"

_PeerJ, doi:10.7717/peerj.17812_

## Round 0.1 · original submission · Minor Revisions

Dear authors, based on the reviewers' reports I reached a minor revisions decision. please address the reviewers' comments. and make sure to carefully proofread the language as well as to assure any image quality standard.

Reviewer 1 ·

Basic reporting

- In the forest plots, when showing the extremely small p-values from the heterogeneity test, please use scientific annotations instead of 0.000.

Experimental design

- Could you provide one or two sentences to provide the detailed exclusion reasons for the five studies shown in Figure 1?
- For publication bias, could you describe what LFK is measuring? How the magnitude of LFK reflects the publication bias. What threshold or criterion was used to determine no publication bias?

Validity of the findings

- Could you discuss some potential hypotheses in terms of why there was a significant heterogeneity in the prevalence of violence across different regions? Was it due to the difference in the prevalence of education and income level? Or were there any other reasons?

Reviewer 2 ·

Basic reporting

1. I recommend you to modify the title and to write as intimate partner violence among reproductive age women during COVID -19 in Ethiopia. And why you select for reproductive age only, why not for all women?

2. In line 137, why do you put the systematic review and meta- analysis in the exclusion criteria, if not remove it please

3. In line 187, rewrite it the study characteristics briefly and in detailed

4. Rewrite it with the discussion part

5. There is one article published in northern Ethiopia, in 2020 (https://doi.org/10.1186/s12978-020-01002-w). why didn’t include it ?

Experimental design

no comment

Validity of the findings

no comment

Additional comments

6. I suggest you to rephrase the English grammar with those who are proficient in international English or contact professional editing service

Annotated reviews are not available for download in order to protect the identity of reviewers who chose to remain anonymous.

---

## Round 0.2 · Minor Revisions

Thank you for resubmitting your manuscript. We appreciate your efforts; however, it is surprising to see that many typos, mistakes, and issues with language usage remain unaddressed. Proofreading the final version before resubmission would have facilitated a smoother review process. While I did not correct your entire document, I have provided some corrections and comments in the attached document based on your tracked version. Please ensure that your manuscript is revised to use proper, academic, and professional English. If needed, PeerJ offers editorial services to assist with this.

Regarding the reviewers' comments:
-"...please clarify why you selected only reproductive age women for your study and not all women. " You should include a refined explanation for inclusion in your manuscript, also in the discussion:
"In this study, we focused specifically on physical, psychological, and sexual intimate partner violence among women of reproductive age during COVID-19 in Ethiopia. We chose this age group for several reasons. Firstly, focusing on the reproductive age group provides insights into the unique challenges and vulnerabilities faced by these women. Secondly, understanding the impact of intimate partner violence on reproductive age women can lead to more effective prevention and intervention strategies that address the intersection of violence and reproductive health. Lastly, studying this specific age group helps raise awareness about the prevalence and impact of intimate partner violence on women's reproductive health and overall well-being, and can highlight more and better interventions that can benefit future generations"... (something of sorts...)
These points should be included both in the background and the discussion sections of your manuscript.

- For line 187, where study characteristics are described, the numbers should be contextualized. Please include the source and methodology of sample collection.

"In this comprehensive study, a total of 4,439 samples were included to assess the prevalence of physical, psychological, and sexual intimate partner violence among women of reproductive age during COVID-19 in Ethiopia. These samples were collected from (....) various regions and demographics within the country registered in reports XYXYXYX. Among these, cases of physical, psychological, and sexual intimate partner violence were reported as 988, 1,233, and 1,020 respectively." (based on a sum from reports xyxyxy??!?)

Please also ensure the revised manuscript accurately reflects these details. While your efforts to redo the analysis and include additional relevant articles are appreciated, it is essential that the manuscript undergoes a thorough revision for clarity and accuracy.

Best regards,

---

## Round 0.3 · accepted · Accept

Dear authors,

Your manuscript retains a considerable amount of typos and language issues, which i think that can be fixed during the proofreading and production stage. I believe that the scientific content of your work is valid and correct. Therefore, I am accepting your work for publication in PeerJ.

Reviewer 1 ·

Basic reporting

The authors have sufficiently addressed my comments.

Experimental design

The authors have sufficiently addressed my comments.

Validity of the findings

The authors have sufficiently addressed my comments.